# The Role of Sustainability in Brand Equity Value in the Financial Sector

**Samer Ajour El Zein [1,*], Carolina Consolacion-Segura [2] and Ruben Huertas-Garcia [3]**

[1]  School of Business Administration, Technical University of Catalonia, Barcelona C. Jordi Girona 1-3, 08034 Barcelona, Spain

[2]  School of Telecommunications Engineering, Technical University of Catalonia, Barcelona C. Jordi Girona 1-3, 08034 Barcelona, Spain; Carolina.consolacion@upc.edu

[3]  Economics and Business Administration Department, University of Barcelona, Avenida diagonal 690, 08034 Barcelona, Spain; rhuertas@ub.edu

*  Correspondence: samer.ajour@upc.edu

**Abstract:** The behavior of firms is changing as new kinds of businesses evolve. In particular, companies are now seeking to optimize their value, especially their intangible value—referred to as brand equity value—which has many behavioral drivers. The analysis of brand equity determinants in the financial sector (e.g., ethical investments, sustainability and firm behavior) has received little attention. The methodology used in this study included the collection of information from publicly listed companies, followed by the execution of a statistical analysis to study the correlations between brand equity values and their determinants. We aimed to close this gap by raising the awareness of the positive impacts of sustainable investments in the financial sector and the need for a managerial implementation model to build a sustainability-oriented brand value. The objective of this research was to examine the relationships between elements such as sustainability scores or diversity measures and firms' brand value. Considering sectoral and regional effects, we observed a positive relationship between environmental and social governance scores and brand equity value.

**Keywords:** sustainability; ethics; brand equity; governance

## 1. Introduction

In recent years we have seen a growing interest for responsible investment, an approach that considers environmental, social and governance (ESG) factors in portfolio selection and management. In 2015, 218 US funds had integrated ESG factors into the investment process. In 2018, this number has increased up to 351, reflecting the growing importance of responsible investment, reaching $161 billion of total assets under management [1]. However, not only United States investors are integrating ESG factors on investment decision but this is a worldwide trend. In fact, global sustainable investment has increased a 67% in the last four years from $18,276 billion in 2014 to $30,683 billion in 2018 in the five major markets [2]. This popularity of sustainable investment may be viewed as investors becoming aware of environmental sustainability, the treatment of companies to their employees and society as a whole, as well as in business policies such as the diversity of the board of directors and ethics business. Nevertheless, investors may not be as altruistic and base their investing choices on sustainable firms because they expect to get better financial returns. In fact, ESG factors may improve a business' image for its stakeholders and engage its clients, boosting brand value.

On the other hand, brands are one of the most strategic assets of a firm, able to get sustainable competitive advantage over competitors. However, companies' financial statements do not include them, so that estimating their values is a hard task. In fact, brand equity may be seen from the consumer

perspective -perception or behavioral value- or the financial perspective -revenue differential between a branded and a generic product. Here, we use brand equity and brand value indistinctly, referring to the financial perspective. The brand value estimation is as proposed by Damodaran [3], where a well-known brand—with customer engagement—can charge a price premium relative to generic brands—without customer engagement. The intuition is the following—firms can charge higher prices for the same products, driving up profit margins and price-sales ratios, as well as firm value. The larger the price premium a firm can charge, the greater the brand value.

The need for this study has arisen from the availability of more modern sustainability data due to increased reporting [4] by public firms and the wide variety of firm valuation methods. We are not the first to describe the relevance of sustainability measures within the business framework. Corporate social responsibility (CSR) efforts have previously had significant impacts on the customer-based brand equity perspective [5] as well as the conceptual model in the business-to-business market, highlighting the importance of the topic from a stakeholder perspective [6]. The analysis of brand equity determinants in the financial sector such as ethical investments, sustainability and firm behavior—being important internal and external sources of brand equity determinants—has so far received little attention. We aimed to close this gap by increasing the awareness of the positive impact of sustainable investments in the financial sector.

In this paper, we go further ethical considerations and we seek to throw additional light on the ESG literature by estimating the impact of ESG investments on brand value. In particular we carry out this analysis on financial sector, which may have found an opportunity to recover its image, reputation and brand value by increasing its concern on social and environmental aspects after its image had sharply been reduced since 2008 financial crisis.

We use an OLS model controlling by region and time effects, what allows us to infer a linear relationship among brand value and ESG factors but not causal effects. Our results suggest that environmental, financial and governance factors are drivers for boosting brand value. That is, the more important are ESG factors for a company, the higher the brand value.

In spite of vast literature on ESG and financial performance, there is a lack of literature on ESG factors' effects on brand value of financial firms. In this study, we close this gap and we find that ESG factors are key to stakeholders by enhancing brand value, what increases competitive advantage of the branded firms relative to generic firms.

The paper is organized as follows. In Section 2, we present a literature review under four broad classifications. In Section 3, we explain how we construct brand value, which are the main variables included in the econometric specification and we describe data statistics. In Section 4, we present our results. Finally, in Section 5, we exhibit the main conclusions and discuss the practical implications and limitations to the study.

## 2. Literature Review

### 2.1. Intangible Assets and Brand Value

Intangible assets are a key factor in the long-term success of any company. As such, their value must be carefully considered. The value of tangible assets is estimated based on future events that are numerically quantified to assign a fair value to each asset [3]. Intangible assets are not easily valued because of their different nature. The difference between an intangible and a tangible asset is their assigned virtual perception. For instance, two investors would assign different values to the same intangible asset because a virtual benefit is delivered that is perceived differently. As the benefit obtained is not physical, the valuation process is more difficult [7]. Considering two companies in the same industry with equal service and other factors, the perception of this extra benefit to the final users gives this brand a different value depending on stockholder and consumer perspectives.

Intangible assets have been under the spotlight due to their growing importance within the business world. Finance, accounting, business strategies and economics have always considered the importance of this category of asset as a fundamental component of a company as a whole. Intangible

assets represent the intellectual capital of a firm as well as its potential growth through innovation, which seems to be the keyword in today's business. Marketing and firm value play roles in creating brand equity value, as appropriate marketing skills and other brand equity determinants affect the shareholder value [8]. Even though this topic is receiving significant attention from practitioners, debate about its features is ongoing, starting with its definition [9]. Some consider intangible assets to be goodwill data. For example, intangible assets include elements such as patents, trademarks, copyrights, brand names or logos that constitute the firm's goodwill [10]. In addition to working capital and tangible assets, intangible assets are classified as a core element of a business enterprise [11]. Therefore, these are the elements that allow the business to operate and can be the primary contributors to a firm's success and competitive advantage [11]. The overall trend in the business world is to conceptualize the day-to-day procedures used to improve performance and increase revenue streams through which companies create value [12].

However, there is no consensus in literature on the meaning and the measuring of a brand. In fact, Winter (1991) explains this discrepancy by stating, "if you ask 10 people to define brand equity, you are likely to get 10 (maybe 11) different answers as what it means". References [13–15] use both terms, brand equity and brand value indistinctly. This terminology difficulty arises because brand equity is more than just a name and a logo [16]. This intangible asset represents an organization's engagement with a customer to deliver what the brand represents in terms of emotional, social and economic benefits. In sum, brand equity may be seen from the consumer perspective—perception or behavioral value—or the financial perspective—revenue differential between a branded and a generic product. Brand equity usually refers to the broad term—including both the consumer and financial perspective—while brand value usually refers to the financial perspective. From this last perspective, Bahar Gidwani (2013) [17] found that sustainability performance and brand value are positively related, what sustains our main hypothesis that ESG factors boost brand image, brand reputation and, hence, brand value.

In this paper, we use brand equity and brand value indistinctly, referring to the financial perspective. We adopt brand value estimation proposed by Damodaran [3], who examined this intangible asset as an incremental cash flow of branded relative to unbranded companies. His model assumptions were built on the premise that the brand name company and a similar generic company are both publicly traded. His proposition is based on the market observations of both companies, which allows a value the difference between the two brand values. Bahar Gidwani (2013) finds that sustainability performance and brand value are positively related.

*2.2. Sustainability Brands and Financial Performance*

Many authors have tried to find the effect of sustainability and social responsibility of firms on financial performance. Corporate Social Responsibility includes a company's social activities, demonstrating the inclusion of social and environmental concerns in business operations. The idea of the only responsibility of a business being to increase its profits dates back to the 1970s [18]. Despite this, companies in the industrial and service sectors were more worried about indirect losses than indirect gains affected by their corporate social responsibility [19]. Since 1978, researchers have noted a correlation between CSR and financial performance [20], which led academics to extend their research in 1985, showing that less-diversified businesses have better corporate social performance [21]. In 2003, the capital market's response relationship to CSR was linked to the amount of information disclosed [22]. As a result of the 2008 financial crisis, considerable research has been conducted on how companies react to external challenges, and large capitalization firms have been reported to have become less responsible [23].

The effects of CSR on corporate financial performance vary across firms and time [24]. Corporate social performance is positively related to a company's reputation [25]. However, in both the banking sector and chemical industry [26], up until 2011, there was no significant relationship between ethical ratings and corporate financial performance. In contrast, CSR has been positively associated with the

firm value of European manufacturing firms [27] in the oil and gas industry [28]. In addition, in recent years, The Conference Board has found an increasing connection between sustainability and brand value [17].

Ameer (2012) finds that companies which attend to ecosystems, societies and environments of the future have higher financial performance compared to those that do not engage in such practices and this superior performance is sustained over time [29]. Good environmental performance [30] is significantly associated with good economic performance and this tends to lead to positive future performance [31] and lower risk exposure, as a result of the social responsibility actions taken [32]. Poor company financial results are generally the result of poor community engagement rather than poor social performance in terms of environmental factors [33]. However, Farooq [34] finds that ESG disclosure is negatively related to firm performance in emerging markets and argues this result by stating that stock market participants can consider ESG investments as unnecessary costs.

### 2.3. Sustainability Brands and the Financial Sector

Sustainability can be defined as meeting human necessities while at the same time preserving the nature or our planet. It is a connection between nature and society [35]. Sustainable science is a field that is trying to examine the correlations between society and resources, how these resources have been used and their limitations and boundaries. It is also trying to address the behavior of the organizations and their responsibilities towards society and nature [36]. In today's business world, sustainability is affecting competitiveness [37]. Executives are very aware that failure on sustainable challenge impacts their organizations in a negative manner [38]. Sustainable strategy became very important on the road map for every organization [39]. Consumers are searching for the sustainable environmental friendly products since concerns about climate change have increased [40]. In order for companies to get a sustainability advantage they need to have green product offerings [41]—sustainable products designed to minimize environmental impacts during its whole life-cycle and waste.

The United Nations (UN) [42] has looked at the reporting of sustainability indicators in the financial sector. Also, private initiatives such as the Asset Owners Disclosure Project can help to promote transparency and, especially if governments promote their use, enable market forces such as reputational impact to take action. The UN is not the only international organization to mention the importance of sustainable investments and indices, as the European Commission [43] recently advised that an increased focus on environmental, social and governance indices during the investment process is necessary. Similarly, Marcel [44] suggested the use of legal and social incentives but also stressed the importance of price incentives to internalize negative externalities on the environment in order to maximise the social welfare.

The financial sector contributes, both positively and negatively, to sustainable development, so there is a need to conduct research in this area to optimize the positive effect [45]. New financial products and social challenges are highly correlated in the banking sector [46]. Despite the important relationship between finance and sustainability and researchers, the need remains to expand the knowledge on the issue of financial management and the concern with sustainable development [47]. This could increase managers' awareness of the relationship between society and the firm when making their decisions [48].

### 2.4. Sustainability Brands and Marketing Strategies

CSR affects the behavior of firms from all sectors and a direct relationship exists between sustainability and marketing strategies. Stakeholders form part of the sustainable scheme by enhancing the added value of a firm [49]. In the industrial sector, there is a positive association between CSR and corporate reputation [50]. For example, there is a conceptual framework in the life insurance industry that shows the impact of CSR on brand equity to be positively related to persuasive advertising effects [51]. In the electronics sector, there is a positive relationship between green characteristics (green satisfaction, green affect, green trust and green brand loyalty) and green brand equity [52].

The incorporation of an ecological method in a brand produces a stronger preference for hedonic attributes.

For this reason, many companies focus on investments in intangible assets and, in particular, in brands and human capital, among others, to ensure the development of a stronger and sustainable image. Thus, they opt for a strategy of converting intangible assets to tangible assets to create the firm's value and place in the market [53]. For the past 30 years, companies have focused on corporate sustainable development and this has become an organizational determinant [54]. This phenomenon has arisen from companies seeking a competitive advantage and trying to become sustainable in parallel with the main business objective, to the point that sustainability can be the profitability tipping point in business. For this reason, sustainability is now a key driver of innovation [55]. The additional benefit of sustainability is that it links social entrepreneurship with economical profitability to the extent of recognizing the social return on investment and triggering the evolution of business strategies [56]. Firms should develop different strategies to achieve a competitive advantage and should focus on asset specificity in determining the multiple uses and purposes of their assets [57]. Since a link exists between strategy and society, a new method was proposed by Porter to link business to societies [58]. For instance, the supply chain sector dealt with this as a business opportunity in 1996 with the introduction of this new scheme in the re-engineering of the structure and management of the supply chain to manage the environment to more effectively use current resources to balance sustainability and profitability. Those changes were meant to represent an investment by the sector, despite being forced by consumers who push producers into developing sustainable products by their desire to use products with minimized environmental effects. This pushed companies to consider the balance between sustainability and pragmatism, which, in turn, affected the brand equity of the whole sector [59].

Businesses have insufficient knowledge about how to see and value CSR. The development of reputation and brand equity require the use of an effective strategy to achieve a competitive advantage and build a company's identity. Thus, a framework needs to be set to identify the contributions of intangible assets based on case studies and to reveal their importance in a sustainable, competitive advantage strategy [60]. A firm's environmental orientation could influence their corporate brand value [61], suggesting that managers should invest wisely in environmental activities, as these investments have an effect on corporate intangible assets [62]. When implementing procedures, managers should consider that company identity can grant a competitive advantage [63] that translates into better performance while still recognizing the importance of the availability of resources [64].

## 3. Model, Methodology and Data

### 3.1. Model

The method for determining brand value (or brand equity) involves considering how much more a consumer is willing to spend on one branded product versus another as well as the fact that there is a relevant branding shareholder value creation link [65,66]. Damodaran [3] examined this intangible asset as an incremental cash flow of branded to unbranded companies. His model assumptions were built on the premise that the brand name company and a similar generic company are both publicly traded. His proposition is based on the market observations of both companies, which allows a value to the difference between the two brand values.

The *Brand Name Value* can be determined as follows:

$$Brand\ Name\ Value = \left[ \left( \frac{EV}{Variable} \right)_{Brand\ Name} - \left( \frac{EV}{Variable} \right)_{Generic\ Brand} \right] * Variable_{Brand\ Name}, \quad (1)$$

where *EV* is the Equity Value. Under the assumption of using *EV/Sales* ratios as multiples for comparison, this would be modified as follows:

$$Brand\ Name\ Value = \left[\left(\frac{EV}{Sales}\right)_{Brand\ Name} - \left(\frac{EV}{Sales}\right)_{Generic\ Brand}\right] * Sales_{Brand\ Name}. \qquad (2)$$

Fernandez [67] underlined a further limitation behind their model (shown in Equation (2)), stating that sales are not identical between the generic brand and the branded company and suggested expressing the following formula to consider the different volumes:

$$Brand\ Name\ Value = \left(\frac{E}{S}\right)_{Brand\ Name} * Sales_{Brand} - \left(\frac{E}{S}\right)_{Generic} * Sales_{Generic},$$

where *E* is the equity calculated by market capitalization and *S* is the sales volume. Therefore, Brand Name Value is the market added value for a branded firm relative to a generic firm.

The control variable Intangibility is an approximation of net intangibles that is computed by

$$Intangible\ Assets = Goodwill/Total\ Assets,$$

which represents the goodwill to assets ratio used to determine what portion of a company's assets are classified as intangible assets relative to its tangible assets. Goodwill is the excess purchase price over the acquiree's book and is considered to be carried on in the new books after the sale of a business as an asset and is eventually written off. The concern here is to determine whether there is any significant relationship between the intangibility and the brand value assigned. In addition, the Return on Assets (ROA) is included as an indicator of how profitable a company is relative to its total assets and the Price-to-Earnings ratio (PER) is a measure of the company's value based on its current share price relative to its per-share earnings.

*3.2. Method*

The dependent variable Brand Value is composed of the ratio of market capitalization to sales, standardized by the sector's generic firm. Despite the limitations on the sales volume [67], we propose the use of various independent variables and test their significance and effect on the brand value. We aim to develop a better brand equity model that considers other significant factors; in particular, sustainability. The first possible weakness considered in this model is the choice of the generic company, as there is difficulty involved in estimating the parameters of the generic product. The choice of the generic company can vary within the same sector, as the ratio of branded to generic companies can vary among sectors, therefore increasing the chance of a hidden arbitrariness in the dependent variable [68]. To reduce this hidden arbitrariness, we propose using the average industry as a proxy for the generic company as the dependent variable of brand equity based on The Bloomberg Industry Classification Systems (BICS) first level of detail [69]. For the independent variables, data were collected and collated using publicly available annual reports from Bloomberg to find approximations of the levels of competitiveness, market share, net intangibility assets, sustainability and transparency and governance factors.

We run a Panel data OLS model regression controlling by region and time effects, which allows us to infer a linear relationship among brand value and ESG factors but not causal effects. We assume that a regression analysis is a statistical procedure to obtain estimates. Causal analysis is not a specific statistical procedure, it can be regression analysis, path analysis or variance analysis. In our paper, the data analysis for research design allows causal conclusions, thus the regression analysis on our data is considered to be a causal analysis [70]. We thought of doing Granger causality to study the econometric relationship that tests whether additional information from the behavioral variables (ESG scores) help explain the brand value. But the independent variables and the brand value variable should be stochastic variables which is not the case. Nevertheless, in the regression analysis this assumption is not necessary (in this case the OLS panel data with dummy variables such as region

controlled by years, there is no need to have stochastic variables). Therefore, the variables could be deterministic, which is the case of the independent variables included in this paper.

Our results suggest that environmental, financial and governance factors are drivers for boosting brand value. That is, the more important are ESG factors for a company, the higher the brand value.

*3.3. Data*

Due to the usage of several variables across 5 years, a panel data/longitudinal dataset was constructed. Tests for multicollinearity among some variables as well as heteroskedasticity, were conducted. Five years of data (2013–2017) were collected from a published annual report of 1100 companies from S&P 500 and EURO 600–Bloomberg. Overall, 1816 observations were collected from a variety of international companies. Our sample thus includes the biggest companies in the the US and European markets. Although, to a varying degree, these markets consist of many small and medium enterprises (SMEs), as publicly traded companies are intensely valued by the market and they more clearly disclose their ESG investments.

Our hypothesis is that the dependent variable, Brand Value, is positively affected by investments in environmental and social governance factors, in addition to other social aspects like the share of women on the board of directors and the proportion of female employees. ESG factors may boost the image and reputation of a firm, with the potential positive effect on customers willing to pay a premium for the branded sustainable product [17]. More female workers and women on the board of directors increases diversity and inclusion, what is clearly correlated with ESG factors—in concrete, with Social factors—so it needs to be included as a control. Also, studies as Shrader (1997) found that firms employing greater percentages of women managers at the general management level experienced a better financial performance in terms of ROS, ROA, ROI and ROE [71]. Since we are working with panel data, we control for yearly and regional effects in order to capture the influence of aggregate trends (time-series) and regional effects that may be correlated with other explanatory variables such as ESG. We include dummy variables for these factors to increase the robustness of the specifications.

Many CSR investment funds have been developed, despite the need for new value creation sources [72] and the recommended enforcement [73,74] of the widely used sustainable reporting instruments and indices. For this reason, the independent variables in this paper is ESG factors, which provides a single company's ESG performance score as well as being based on third-party ESG scores; the quality score of the Institutional Shareholder Services (ISS), which is the world's leading provider of corporate governance and responsible investment solutions and the collective voice of shareholders; and the SR (Sustainalytics Rank), provided by a global investment firm that specializes in sustainability research and analysis, to show the sustainability of a company. To tackle the company behavior, the Company Disclosure Performance score (CDP) was added as an index to measure the transparency of companies, followed by two variables: the number of women on the board and the number of female employees. Finally, we also include a categorical variable for whether the firm is from the US or the EUR market and include time in the main regression as control. A summary of the main variables used is provided in Table 1 and the main statistics related to the financial and social factors are displayed in Table 2.

ESG has been positively linked to corporate financial performance across a wide range of more than 2000 research articles [75]. It is important to use several of these variables, as there are important differences among ESG rankings, so the use of just one might lead to biased results.

In Figure 1, we display the density of the estimated Brand Value in our data set for every year. Interestingly, this shows a trend of increasing dispersion, with more firms having even more negative brand value and large, positive outliers.

**Table 1.** Main variables.

| Latent Variables | | Observable Variables |
| --- | --- | --- |
| Financial situation | Intangibility | Measured intangible assets |
| | ROA | Return on assets |
| | PER | Price–earnings ratio |
| | Index Growth | Average growth of relevant index |
| Social factors | ESG score | Environmental and Social Governance index |
| | Women Directors | Share of female directors |
| | Women Employed | Share of women employed |
| Unobserved factors | Region | US/EUR |
| | Time | Year variable |

**Table 2.** Summary of the statistics.

| | Mean | Median | Std. Dev | Max | Min |
| --- | --- | --- | --- | --- | --- |
| Brand Value | 157.5 | −8989.6 | 47,775.99 | 733,090.4 | −97,977.6 |
| Intangibility | 0.077 | 0.037 | 0.001 | 0.799 | −0.009 |
| ROA | 5.930 | 4.881 | 0.09 | 235.4 | −70.4 |
| PER | 1631 | 535 | 46.8 | 141,828 | −35,206 |
| ESG | 36.861 | 37.191 | 0.151 | 78.512 | 3.509 |
| Women Directors | 74.81 | 80.00 | 0.179 | 100.00 | 0.00 |
| Women Employed | 36.94 | 35.00 | 0.160 | 84.7 | 6.0 |

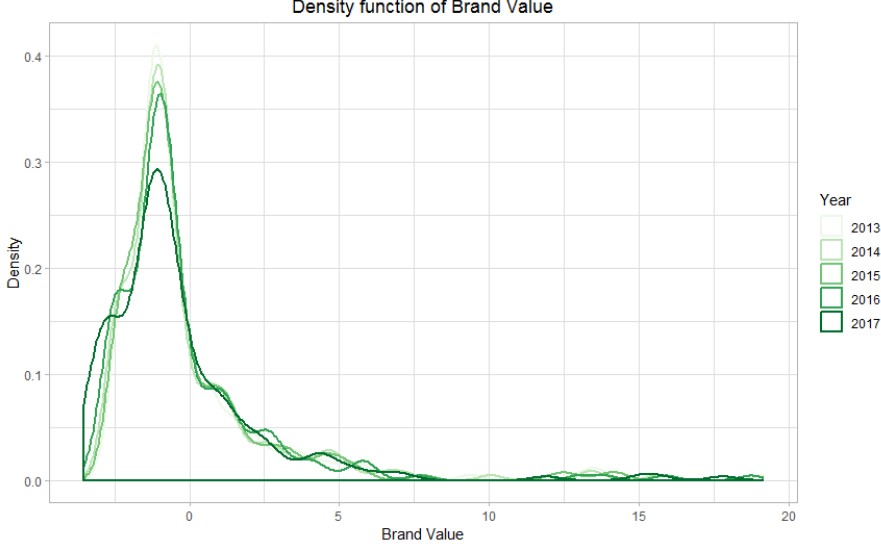

**Figure 1.** Annual density of Brand Value.

In Figure 2, we display each firm's average brand value, sector and either the average Sustainalytics index or the average Environmental and Social Governance score, in both cases with a generalized additive model with integrated smoothness displaying the trend. As can be seen, across different indices and levels, higher values of sustainability tend to be correlated with higher average brand values.

However, as can be seen in this figure, there is wide variability in brand values when considering companies from all sectors based on the Bloomberg BICS classification. Subsequently, we conducted a more detailed analysis of brand value, particularly in the financial sector.

In Figure 3, we display the same study for only the financial sector. The results are consistent within this sector, as both indices had a similarly positive, albeit not linear, relationship with brand value.

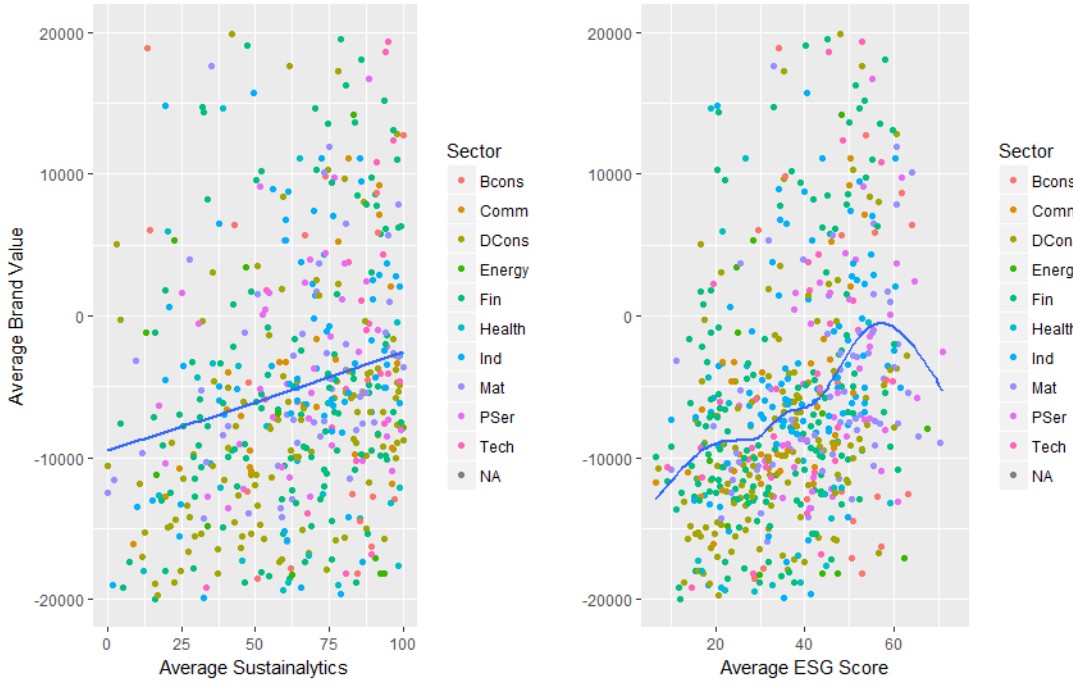

**Figure 2.** Brand value and Sustainability Indices.

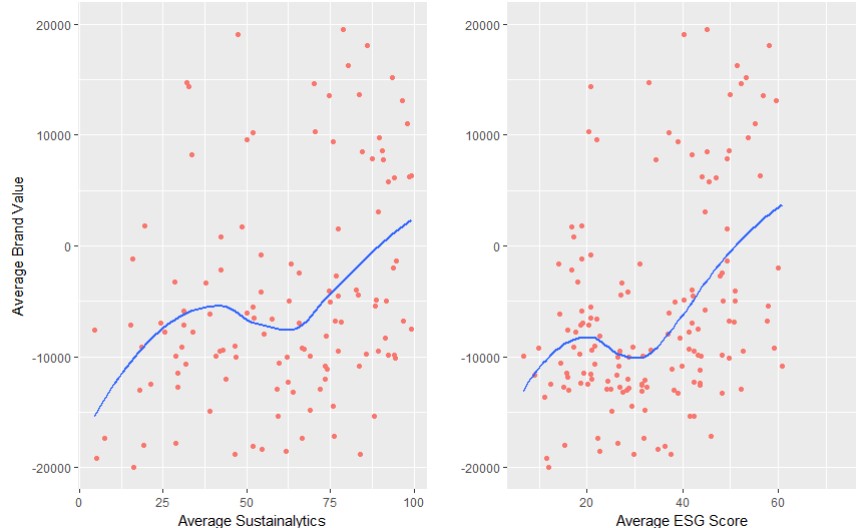

**Figure 3.** Brand value and Sustainability Indices for the Financial Sector.

To study the evolution of this relationship between Brand Value and the Environmental and Social Governance score, Figure 4 shows a smooth trend linking both variables for every year of our sample. The trend was constantly positive over time and even displayed higher steepness in the last two years. This could imply that in more recent years, higher ESG scores were being more positively received by the market.

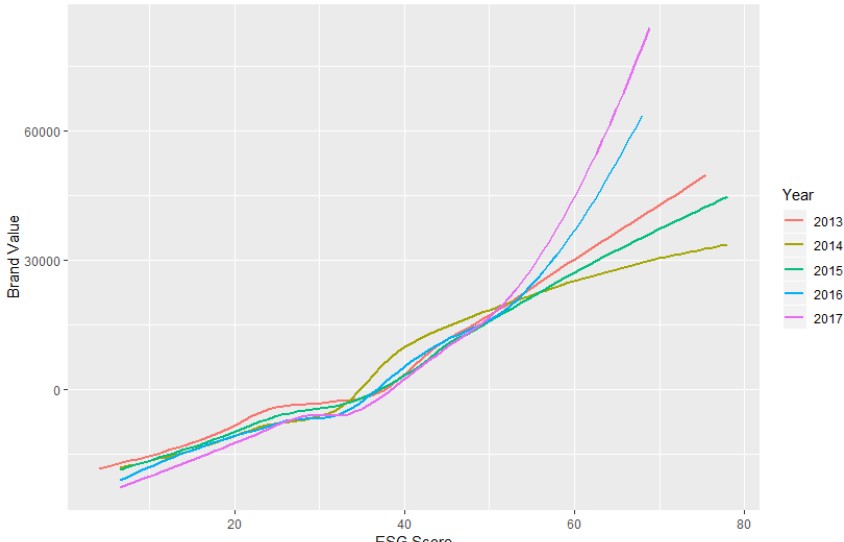

**Figure 4.** Evolution of the relationship between the environmental social and governance (ESG) score and Brand Value.

## 4. Empirical Results

Our findings shed some initial light on how brand equity is affected by environmental and social governance reporting of the underlying company in the financial sector. Results from the regression and paired-sample *t*-test methods show with a very highly significant *p*-value that a higher ESG score for a given company corresponds to a higher brand equity value (Table 3 ).

**Table 3.** Regression Results

|  | *Dependent Variable:* |
| --- | --- |
|  | **Brand Value** |
| Intangibility | 16,732.2 *<br>(8978.9) |
| ROA | −205.64 *<br>(81.6) |
| PER | 4.2 ***<br>(0.1) |
| ESG score | 648.2 ***<br>(81.0) |
| Women Directors | 103.1 *<br>(52.8) |
| Women Employed | 8.1<br>(56.9) |
| Index growth | 38,296.8<br>(3085.9) |
| Region (U.S.) | 11,290.0 ***<br>(3085.9) |
| Constant | −44,198.7 ***<br>(7497.0) |
| Observations | 2467 |
| $R^2$ | 0.281 |
| Adjusted $R^2$ | 0.278 |
| F-Statistic | 80.101 *** (df = 12; 512) |

Note: * $p < 0.1$; ** $p < 0.05$; *** $p < 0.01$.

To test for multicollinearity, we used the variance inflation factor test (VIF), which compares the variance of the model with several factors with the model with one term alone. The results, in Table 4 show that the variances of the estimated coefficient of all variables were moderately inflated, while the VIF values of all the other variables were below 10, indicating low correlarity among the independent variables and that the multicollinearity does not pose a problem for our explicative model used (Appendix A).

**Table 4.** Variance inflation factor (VIF) test.

| Variable | Test Statistic |
|---|---|
| Intangibility | 1.012 |
| ROA | 1.069 |
| PER | 1.073 |
| ESG score | 1.086 |
| Women Directors | 1.349 |
| Women Employed | 1.047 |
| Index Growth | 3.732 |
| Region | 2.419 |
| Year | 3.396 |

To validate the appropriateness of the model we are using, we perform residual analysis (difference between the predicted response and the actual response) and examine residual plots to evaluate how well the model fits the data and that the data meet the assumptions of the model [76]. Residuals are plotted to understand whether the assumptions which have gone in building a linear model hold true or not.

The residual plot for the Brand Value dependent value with each of the independent variables shows that most of the model validation centers around the residuals (essentially the distance of the data points from the fitted regression line) validating homoscedasticity that means that the residuals are equally distributed across the regression line, that is, above and below the regression line and the variance of the residuals should be the same for all predicted scores along the regression line. This accepts the assumption of validating the appropriateness of the model we are using.

To test for heteroskedasticity in the linear regression model to check whether the variance of the errors from the regression was dependent on the values of the independent variables, we used the Breusch–Pagan (BP) test, which indicates whether the variance of the errors depends on the values of the independent variables. The results, displayed in Table 5 showed a very low $p$-value; thus, the null hypothesis of homoskedasticity was rejected and heteroskedasticity was assumed.

**Table 5.** Breusch–Pagan (BP) test.

| Variable | Model |
|---|---|
| Statistic | 689.72 |
| Degrees of freedom | 12 |
| $p$-value | 0 |

As seen in the main Table, Table 3, the dependent variable, Brand Value, was positively affected by the ESG score. Consistent with the higher average Brand Value, the dummy variable for the United States (US) region was also significantly positive. Although tangible and intangible attributes and both are found to be important contributors to brand equity and brand choice [77], Intangibility variable was as well positively significant. The share of Women Directors, being a business imperative [78] were also positively correlated with the Brand equity Value, albeit less significantly. Even though the share of female directors did positively affect the brand value, the share of female employees had no direct implication on brand value and was not significant. Although statistically it makes

sense to eliminate effects that are not serving a purpose, but this insignificant effect has a purpose in highlighting that, even though diversity affects positively on business performance [79], we found out that in the financial sector, gender diversity does not affect brand vale . A similar result was found when including other sustainability indices, such as Sustainalytics, in the financial sector . Since sustainability ratings are a challenge to financial firms [80], the importance of this result advises firms of the importance of Sustainalytics and the possible future positive effect on brand vale. The ESG score did not lose significance but the new addition simply generated noise and was not significant.

Finding the impact of sustainable investments on financial firms' Brand Value is considered difficult due to nature of the valuation methods of intangible assets, as mentioned by Salinas [81]. Since low multicollinearity exists for the independent variables in our model, already discussed in the previous paragraph, then we can interpret the effect of the independent variable on the dependent variable by considering the coefficients [82]. The positive significant coefficients are ESG score index, Intangibility, Price to Earnings ratio, share of female directors. So an increase of any of these variables would increase the Brand Value. More precisely, our results suggest that an ESG score index increase of one unit would boost Brand Value in 648.2 million dollars, on average. This result indicates that financial firms will end up improving their Brand Value by further investments in sustainable investments, thus enabling those investments to be the preferred investment focus in the financial sector [83]. Also, for each additional unit intangibility, we can expect an average increment of Brand Value of 16,732.2 million dollars, that prove the contribution of human capital as an intangible asset to Brand Value [84]. In line with standard accounting assumptions, the price to earnings ratio (PER) has a positive effect on our dependent variable, which is in line with increasing number of investors using PER ratios to make decisions [85] and furthermore, we can observe that for each additional unit of Price to Earnings ratio, the Brand Value is expected to increase by an average of 4 million dollars, a motive to guide the organization focus on increasing shareholder value [86] in the financial sector. An additional unit of participation of female director would increase the brand value on 103 million dollars, on average, that provides implications for future research regarding the effectiveness of female board of directors towards firm performance and Brand Value in the firm sector. However, the Return on Assets (ROA) ratio has a negative, albeit less-significant, effect. Robbin [87] referred to the negative relationship between brand value and return of asset in big capitalization firms that coincides with the firms in our data set in the financial sector, what could explain the negative effect. Despite the limitations of valuing brands [88] and our proposed scheme for identifying brand value drivers, that is, the parameters influencing the brand's value, our main challenge in this paper is raising awareness of this positive impact between social drivers and Brand Value. Knowing that social sector is attracting companies in order to identify opportunities for business innovation [89], there is still a need to implement those models both supported by academics and applicable by practitioners in the financial sector to ensure a greener and more sustainable sector.

We also included the average annual growth rate of exchange-traded funds (ETFs) to track the S&P500 and the Eurostoxx 600. In this way, we controlled for possible effects in brand value unaccounted for by the model, such as a slow-moving global trends in stock prices and brand values. However, this indicator did not appear to be significantly related to our measure of Brand Value. The model explained 28% of the variability, as seen through the adjusted $R^2$ and the F-statistic allowed us to strongly reject the possibility of the independent variables' coefficients being zero.

## 5. Conclusions, Managerial Implications, limitations and Further Research

Brands bring awareness to users and allow them to remember a particular product or service [90]. Due to the need to develop awareness of the sustainability concept that is already in place [91], a responsible business guide could contribute to obtaining a better brand equity value. Our suggestions include not only investing more and trying to obtain a higher ESG score but also disclosing those investments and promoting what the company does. The results show, with a reasonable significance level, that the more sustainable a company is, the higher their brand equity value is. In addition,

a more gender-diverse board of directors could positively influence the brand value of a company in the financial sector. As already mentioned, cooperation through reporting to the UN and to private entities that publish such indices should be enhanced. Drawing the line from all of this statistical information, one idea can clearly be underlined: environmental, financial and governance factors are drivers for boosting brand value. That is, the more important are ESG factors for a company, the higher the brand value. This helps raise awareness to management and investors, together to a single goal to draw a distinct image in the consumer's mind with a more sustainable Brand. Differentiation is an inevitable part of brand management, which can be done by positioning and integrated marketing communication [92]. Brand was initially used to differentiate a group of products from that of others [93]; but nowadays, brands are used by consumers to differentiate them within society [94]. It has become a very much integrated in the business models; and consumers have a voice in distinguishing service quality in all sectors [95] and in the financial sector particularly [96] playing a key role for managers to be aware of the reasons and consequences of why customers stay [97] and thus plan for a service quality in an integrative approach [98].

These results also affect the perspectives of the end users, investors or fund managers, as higher ESG scores might signal future long-term gains in brand value that have only recently been captured by the market and included in the price. This relationship could foster a virtuous circle in which companies with green investments attract [99] more capital and are able to grow and invest more. Investors that are able to show metrics on the sustainability of their portfolios can use those metrics as added value that distinguishes them from other fund managers [100]. In addition, due to the interlinkages between the financial sector and the rest of the sectors of the economy, the effects on brand value can spread to other firms and new ways of reporting information [101] and new channels of social investments can be achieved through the classical banking activities of financial intermediaries integrating the behavioral factors we raised awareness in this paper and whose implementation is crucial to achieve a green brand [102].

Our research adds extra questions regarding firms' reporting of environmental and social aspects [103]. This study was limited by the availability of these data and by the complexity of the estimation of brand value [104]. We studied the main trends through the main indexes of sustainability and an estimation method while attempting to reduce analyst bias [105]. However, further research is needed to increase the robustness of the results and contrast them with new data-sets and estimates [106]. In addition, due to data availability, we focused on big firms in the US and EU markets; however, SMEs might be driving their brand value through their social investments even more so than big corporations. Further research should focus on possible nonlinear effects. For example, as seen in Figures 1–3, the relationship between ESG score and brand value, although positive, was not constant and varied over time. Tools such as SPSS softwares solution adopted for SMEs using digital marketing tools to managing brand equity [107] could be a further research for all firms in the financial sector that seek a continuous sustainable trend. Thus further managerial implications on a practical level with an integrated model that takes into account the social, environmental and economic performance for the creation of sustainability-oriented brand value in the financial sector is needed. Doing that is not an easy task; however, the results obtained constitute a small but significant first step by raising awareness of its importance. This first step can provide a guidance starting point for those the financial firms that want to improve their business models and follow the path of growth and sustainability by managing their brand equity for a long run approach.

**Author Contributions:** All authors contributed equally to the conceptualization, methodology, software, validation, data analysis and execution, writing—review and editing, visualization, and supervision of this paper at all stages. All authors have read and agreed to the published version of the manuscript.

**Funding:** This research received no external funding.

**Acknowledgments:** We thank all colleagues that granted support, and acknowledge support received from Technical University of Catalonia and University of Barcelona.

**Conflicts of Interest:** The authors declare no conflicts of interest.

## Appendix A. Model Validation Graphs

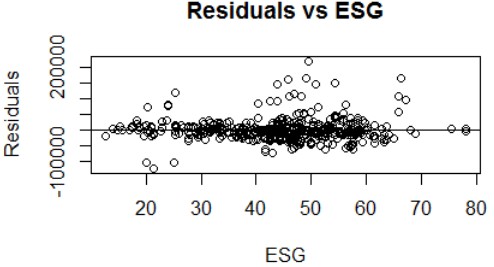

**Figure A1.** Residuals vs Directors.

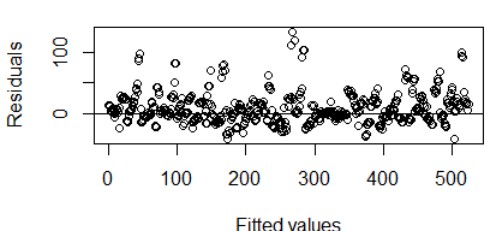

**Figure A2.** Residuals vs ESG.

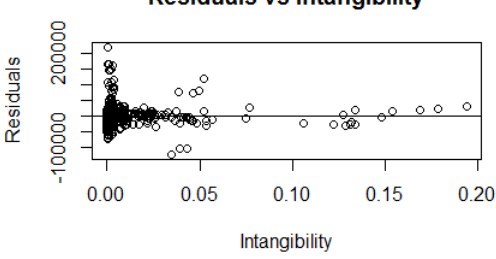

**Figure A3.** Residuals vs Fitted.

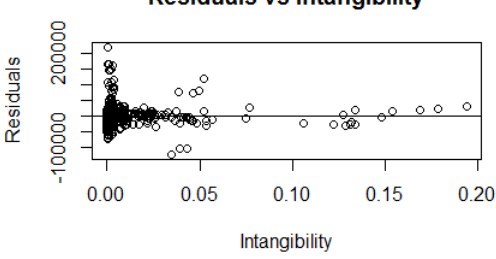

**Figure A4.** Residuals vs Intangibility.

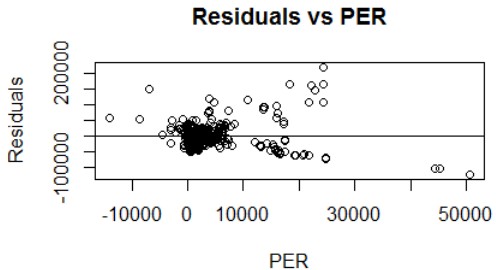

**Figure A5.** Residuals vs PER.

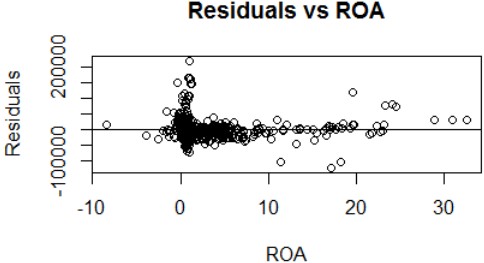

**Figure A6.** Residuals vs ROA.

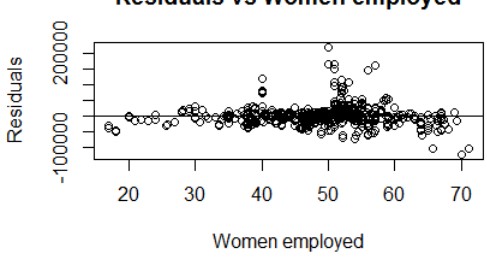

**Figure A7.** Residuals vs Women Employed.

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
