# Peer review of "The Role of Sustainability in Brand Equity Value in the Financial Sector"

_sustainability, doi:10.3390/su12010254_

Round 1
Reviewer 1 Report
Referee report on the paper
The Role of Sustainability in Brand Equity Value in
the Financial Sector
Samer Ajour 1,*, Carolina Consolación 2 and Ruben Huertas 3
OVERALL EVALUATION
The paper deals with a crucial and well established issue in the CSR literature: the nexus between ESG scores and corporate performance. The authors evaluate it using as dependent variable the ratio between market value and sales and define this variable as brand value. I explain below why I’m not convinced about the definition of the dependent variable. Another main problem of the paper is the lack of an identification strategy
SPECIFIC POINTS
The main problem of the paper on my opinion concerns the definition of the dependent variable. It is quite difficult to disentangle brand value from other values. The way brand value is operationalized (market cap/sales) may simply reflect future expected sales. And there are many reasons why sales can grow in the future not all related to the value of the brand. What is brand value ? A sustainable competitive advantage based on monopoly power, on patents, on technological advancements, on organizational efficiency ? Or just something related to the force of marketing strategies ?
A company may have different brands inherited from various mergers and acquisition but what the author measures at the nominator is market value not brand value. If a company has more brands the two (market value and brand value) do not coincide. If we have in mind the fundamental approach to market value (the sum of the expected discounted cash flows) what the dependent variable actually measure is future sales growth (or better cash flow growth) perspectives in a similar way to what a P/E indicator measures.
Endogeneity problems affect the nexus between ESG and brand value. It may well be that more successful companies, companies that have a competitive advantage, have more room to use financial resources for ESG investment. The identification problem should at least be discussed in the paper .What the authors find is just correlation, not causality
There is no idea in the paper of the economic significance of the main result (the effect of ESG on brand value). It is important to give an idea to the reader of the magnitude of the effect
When the main variables of the econometric specification are presented in Table 1 it seems that region and year fixed effects are there. When results are presented there is no trace of them. Their inclusion as regressors should at least be acknowledged in table legend The variable Region (US) is obscure. I hope the authors have not transformed dummy variable separate for each region into a unique variable The literature on the nexus between CSR and corporate performance is immense. The synthesis provided by the author that identifies clear cut results according to different industries is not a good synthesis of it As is well known ESG scores are the sum or average of performance in three well distinct domains (environment, social and governance). It should be nice to have a breakdown of the effect with an additional estimates where the ESG variable is divided into three variables measuring separately the three dimensions
MINOR POINTS
Figure 1 is odd. If there are arrows from the three drivers of brand value there is no need to write “relation” on them. The figure is not commented in the text (in the sense that rationales on the presumed causal links are not provided). The nexus between market cap and brand value measured with a unidirectional arrow seems to indicate a causal relationship from the first to the second driver. Actually it is higher brand value that generates higher market capitalization and it is much harder to imagine a causal relationship in the opposite directionThe authors mention at beginning of section two that banks are less heavily regulated than non financial companies. The claim is not substantiated and seems to go opposite to what we observe.
It would be convenient to mention in the introduction how the empirical analysis is carried out and what are the main findings in order to give soon to the reader a synthesis of main results
Author Response
Please find attached the responses.

Reviewer 2 Report
1. The article have widths of table 3, table 4, and table 5 that are too narrow.
2. The research methodology is too simple. It is only to use SPSS to analyze the collecting data. I suggest the authors to employ a creative research methodology to analyze the data. The results could be showed deeply.
3. English proficiency need to be enhanced on the article content.
Author Response
Please find attached the responses.

Reviewer 3 Report
The theme of the study is of interest to scholars and marketers. Initially, this study tried to provide better understanding on how sustainability can impact on brand equity value as financial sector. However, unfortunately, the central ideas are not examined in this research and overall the paper was not well-organized logically. There are major weakness in the conceptual development, hypotheses development, research design, discussion/implications in this manuscript.
Introduction
The introduction seems rushed and the ideas presented not fully fleshed out. Explain to the reader what the construct is, what research exists, and why the construction is important to the purpose of the study. This paper needs to have a strong justification. How is this study different from other previous studies?
In order to provide the reader with a better understanding of the overall research framework of this paper, Figure 1 should be more sophisticated and moved at the end of literature review.
Literature Review
There is lack of conceptual theory and conceptual framework which can guide and support this study. Also, author(s) ought to include a strong theoretical underpinning for each of your hypotheses in order to test your assumption. Under 3.3 Data section, author(s) stated “We hypothesized that our dependent variable, Brand Value, is positively affected by investments in environmental and social governance factors, in addition to other social aspects like the share of women on the board of directors and the proportion of female employees”. However, there were no hypotheses proposed in the literature review.
The literature review could be stronger, especially in terms of brand equity. When a new concept is brought in to explain or predict an outcome it has to be very well placed and justified with detailed review, linking it to existing frameworks and showing precisely how it adds value/contribution to the existing theory. I am not convinced of the contribution of this paper in its current version.
Prior to using the term, a clear definition of the term should take precedence (e.g., sustainability brands, green brand equity).
Also, confusion of terminology: brand value and brand equity value. Please specify this and clarify in the paper as well. Brand equity should be defined or described upfront as this seems to be the key concept used in the paper.
Method:
How intangibility was measured in your study? Why did you include women directors, women employed, and regions as independent variables to do regression test? It was not stated about these variables. Also no discussion about these in the results and/or implications.
Results
The results showed the Return on Assets (ROA) ratio had a negative impact on brand value. Author(s) did not discuss about it. How could you interpret it? Author(s) need to discuss why this results showed in the paper.
I am confused over how the results are reported.
Discussion/Implications
There is lack of discussion about results. Also, one of the biggest problems in this study is that it is very hard to understand final goal of this study. What are implications for this study? Researcher(s) need to provide more specific practical implications and more details. It is very hard for the reader to see the purpose of this study from the conclusion.
The theoretical contributions and future research avenues are extremely underdeveloped. Managerial implications and limitations are not even discussed.
Author Response
Please find attached the responses.

Round 2
Reviewer 1 Report
Dear authors and editors, I'm still not satisfied about the author 's answer at point 3 Point 3: There is no idea in the paper of the economic significance of the main result (the effect of ESG on brand value). It is important to give an idea to the reader of the magnitude of the effect what I asked here was a quantification of the effect. We know that there is a significant relationship between the main regressor (ESG) and the dependent variable but measuring the economic significance means calculating how much change an X-increase of the regressors produces on the dependent variable I therefore ask to author to revise again this point.Ok for the rest
Author Response
Please find attached the response. Thank you for your comment. It helped and guided us improve this section mentioned.

Reviewer 2 Report
How to employ the equation of Brand Name Value to analyze the data in this article? Please explain clearly. These tables format in this article need to follow APA style or the Sustainability format in order to enhance the article quality. Figure 1,2,3,4. Annual density of Brand Value does not fit correctly the format.
Author Response

(The authors gave the same response as above.)
